# Before Is Better: Innovative Multidisciplinary Preconception Care in Different Clinical Contexts

**DOI:** 10.3390/jcm12196352

**Published:** 2023-10-03

**Authors:** Martina Cristodoro, Marinella Dell’Avanzo, Matilda Ghio, Faustina Lalatta, Walter Vena, Andrea Lania, Laura Sacchi, Maria Bravo, Alessandro Bulfoni, Nicoletta Di Simone, Annalisa Inversetti

**Affiliations:** 1Department of Biomedical Sciences, Humanitas University, Via Rita Levi Montalcini 4, 20072 Pieve Emanuele, Italy; 2IRCCS Humanitas Research Hospital, Via Manzoni 56, 20089 Milan, Italy; 3Division of Obstetrics and Gynecology, Humanitas San Pio X Hospital, 20159 Milan, Italy; 4Diabetes Center, Humanitas Gavazzeni Institute, Via M. Gavazzeni 21, 24100 Bergamo, Italy

**Keywords:** preconception care, adverse obstetrical events, reproduction, multidisciplinary team

## Abstract

Context: Implementation of pre-conception care units is still very limited in Italy. Nowadays, the population’s awareness of the reproductive risks that can be reduced or prevented is very low. Purpose and main findings: We presented a new personalized multidisciplinary model of preconception care aimed at identifying and possibly reducing adverse reproductive events. We analyzed three cohorts of population: couples from the general population, infertile or subfertile couples, and couples with a previous history of adverse reproductive events. The proposal involves a deep investigation regarding family history, the personal histories of both partners, and reproductive history. Principal conclusions: Preconception care is still neglected in Italy and under-evaluated by clinicians involved in natural or in vitro reproduction. Adequate preconception counseling will improve maternal and fetal obstetrical outcomes.

## 1. Purpose

It is well known that preconception interventions, and specifically the promotion of optimal preconception maternal and paternal health status, reduce the risk of adverse reproductive events. Despite this, in Italy, preconception counseling is still overlooked.

The aim of our article is to describe a multidisciplinary model of preconception care for three cohorts of patients:The general population looking for a pregnancy for the first time;Infertile and subfertile couples;Couples with a previous history of adverse reproductive events.

Our purpose is to provide clinicians with guidelines to offer effective preconception care.

Our article gives a new insight through a multidisciplinary, personalized approach to recognizing red flags that could expose the woman and her partner to preventable adverse reproductive events (stillbirth, recurrent pregnancy loss, and preterm birth).

The main topics are summarized in Table 1.

## 2. Model of Preconception Care in the General Population

### 2.1. Family History

Family history is an important risk factor for many chronic diseases (cardiovascular diseases, type 2 diabetes, and cancers). It reflects genetic and environmental factors.

Drawing up a family tree or pedigree is the best way to collect genetic information.

Consanguinity should be directly asked about because it can increase the risk of autosomal recessive disorders.

### 2.2. Personal History

A personal history investigation should focus on potential women’s chronic disorders (hypertensive disorders, diabetes, autoimmune diseases, epilepsy, and obesity) and genetic diseases.

Focusing on reproductive history, the investigation should include menstrual cycle characteristics, eventual dysmenorrhea, sexual intercourse frequency, eventual dyspareunia, and past gynecological surgery. Counseling before pregnancy should also focus on eventual thyroid dysfunction or hyperandrogenism. It is also relevant to ask if there is a familiar history of early menopause.

Regarding lifestyle factors, job and sport activity, smoking, alcohol, eventual addiction, or use of recreational drugs should be investigated.

Concerning nutritional status, an adequate body weight is fundamental before a pregnancy to have an optimal hormonal balance to promote ovulation and consequently pregnancy and to minimize all the possible risks that can occur to the mother (particularly gestational diabetes, preeclampsia, miscarriage, and preterm delivery) and to the unborn child (increased risk of neonatal adiposity and childhood overweight or obesity).

The standard approach includes the calculation of the body mass index (BMI), expressed as the ratio between the weight in kilograms and the square of the body height in meters (Kg/m^2^): below 18.5, it can be defined as “underweight”; between 18.5 and 24.9, it shows “normal weight”; above 25, it indicates “overweight”; and above 30, it shows “obesity”. The two extremes require a specific intervention.

Several studies have demonstrated that a higher maternal pre-pregnancy BMI is associated with an increased risk of adverse obstetrical outcomes (gestational diabetes, preterm birth, hypertensive disorder, fetal death, stillbirth, and macrosomia) [1]. The same studies showed that even minimal weight loss has positive effects on fertility, increasing ovulatory cycles and reducing possible complications during pregnancy.

Counterintuitively, a lower maternal pre-pregnancy BMI also increases the risk of adverse obstetrical outcomes. In 2022, Nakanishi et al. found a correlation between the severity of low pre-pregnancy BMI, low birth weight, and preterm birth [2]. Tang et al. studied a cohort of 668,956 women. In underweight women (BMI < 18.5 kg/m^2^), there was an increased risk of preterm birth and a small risk for gestational age. In overweight women (BMI: 24–27.9 kg/m^2^), there was an increased risk of high gestational age, primary caesarean delivery, and stillbirth. In obese women (BMI > 28 kg/m^2^), there was an increased risk of preterm birth, high gestational age, and primary caesarean delivery [3]. Summarizing, altered maternal pre-pregnancy BMI is linked with a higher probability of adverse obstetrical and perinatal outcomes.

### 2.3. Andrological Assessment 

Concerning men’s perception of health status, in particular concerning fertility and preconception care, it has been described as having a lower level of awareness and attention to the topic compared to women’s health status [4]. To the authors’ best knowledge, no international guideline or consensus was developed to address preconception male health. No study to date has focused on the possible benefits of such a practice on the couple’s health and conceptional outcomes.

According to the CDC [5], male partners should receive appropriate medical advice before starting to plan a pregnancy. 

First, an appropriate evaluation should be aimed at investigating past medical history and the presence of relevant morbidities, past or present exposure to potentially interfering treatments (e.g., chemotherapy, neuroleptic), environmental exposures, and the presence of previous or even active genital tract infections with potential impact on the reproductive tract.

Second, the presence of risky behaviors should be addressed and strongly discouraged, at least during pregnancy-seeking time: both cigarette smoking [6] and alcohol consumption [7] have been proven to have detrimental effects on seminal fluid parameters. Moreover, among the so-called image- and performance-enhancing drugs (IPED), which are becoming every day more accepted and easily obtainable in the modern lifestyle, a wide variety of anabolic steroids are spreading out, with a possible long-lasting impact on male gonadal and reproductive function [8], so their use should be strongly discouraged. 

Further, male overweight (BMI > 25 kg/m^2^) is associated with lower testosterone levels, sperm count, and motility [9]. 

Last but not least, mental health should be assessed, as many psychiatric conditions can have a negative impact on both future mother and child health outcomes [10]. Specifically, some researchers found that the father’s mental health may influence spermiogenesis, the development from the zygote to the embryo, embryo implantation, and germ layer differentiation. Depression or anxiety have a role in preconception via sperm epigenetic alterations [11,12]. 

### 2.4. Supplementation

Folate (vitamin B9) is an important cofactor for DNA synthesis, methylation, and replication [13]. Folate requirements are higher during pregnancy because of the rapid fetal and placental growth [14]. Some food products have higher levels of folate, such as dark green leafy vegetables, eggs, legumes, offal meats, and fruits. However, it is difficult to achieve an adequate intake of folate only through diet. Since 2004, Italy has established the Italian Network for Folic Acid Promotion. It is recommended to consume 0.4 mg/d of folic acid for at least one month before conception and for the first trimester. The supplementation of 0,4 mg of folic acid ensures an adequate level of folate during organogenesis [15]. In 2015, the World Health Organization published guidelines for optimal serum folate concentrations and red blood cell folate for the prevention of neural tube defects: the red cell folate concentrations should be above 906 nmol/L (greater than 400 ng/mL) [16]. According to Dante et al. [17], in Italy, only 20–25% of women started folic acid (FA) supplementation before pregnancy. It has been reported that Italian women with higher education, primiparous women, and women with children affected by congenital defects pay more attention to the importance of FA supplementation [18]. 

According to the literature, periconceptional folic acid supplementation prevents different types of birth defects, such as neural tube defects, cardiac defects, oral facial clefts, cleft palates, and limb reduction defects [19]. 

Since 1981, first Laurence [20] and then other researchers have highlighted the importance of periconceptional supplementation with folic acid in the reduction of neural tube defects. Neural tube defects include different anomalies such as spina bifida, anencephaly, and encephalocele [21]. In several countries around the world, folic acid led to a reduction of up to 70% in the prevalence of neural tube defects [22]. 

Additionally, maternal periconceptional folic acid prevents the risk of congenital heart defects (CHD) [23]. CHD includes congenital anomalies, such as conotruncal defects and coarctation of the aorta [24]. In different studies, it was demonstrated that periconceptional folic acid is linked to a reduction of 25–50% of CHD [25].

Regarding orofacial clefts (OFC), considerable studies have shown that folate deficiency may increase the risk of OFC. Specifically, multivitamins containing folic acid had a more pronounced effect than folic acid alone, probably because vitamin B1 and B6 could strengthen the efficacy of folic acid [26]. In different studies, the supplementation of folic acid results in a reduction of approximately 50% of OFC [27]. 

Lastly, evidence suggests that inadequate folate concentration is linked to an increased risk of limb reduction defects. In detail, a reduction of 30–40% of upper limb defects was observed in women who took periconceptional folic acid [28]. 

The supplementation of folic acid reduces the risk of congenital defects, but it is also associated with other aspects of pregnancy. Folate deficiency may cause epigenetic modifications that can alter placental and fetal growth. Therefore, as demonstrated by earlier studies, folic acid can reduce the risk of SGA [29]. Several studies have demonstrated that preconception folic acid is linked to a reduction of 60% of SGA birth risk [30]. 

Recently, it was found that supplementation with multivitamins, such as folic acid, vitamin B6, and vitamin B12, also has an important role in the prevention of gestational hypertension and gestational diabetes. In fact, vitamin B6 and B12 may have a synergistic effect with folic acid in reducing the risk of insulin resistance and preeclampsia [31,32]. 

Finally, folic acid also plays a role in spontaneous abortion (SA) [33]. Fetal chromosomal abnormalities represent 50% of the causes of SA. In terms of percentages, folic acid supplementation reduced the risk of SA by 49% because of its influence on meiosis [34]. 

Regarding mineral supplementation, it is already known that maternal iodine intake has a beneficial role in maternal and fetal outcomes [35]. In the preconception period and during pregnancy, thyroid hormone production is increased, as are, consequently, iodine requirements [36]. If iodine supplementation is not adequate, the production of thyroid hormone may decrease, leading to an increased risk of maternal and fetal hypothyroidism, pregnancy loss, and impaired child development [37]. The World Health Organization (WHO) recommends a daily iodine intake of 250 micrograms for pregnant women [38]. 

Another important mineral to consider is iron. Iron deficiency anemia affects 30–50% of pregnant women [39]. WHO recommends a daily iron intake of 30–60 mg for pregnant women to improve birth outcomes [40].

Several studies highlighted the importance of an adequate intake of other minerals and vitamins, such as zinc [41], calcium [42], selenium [43], vitamin A [44], and vitamin D [45]. Preconception counseling should emphasize the importance of dietary changes and micronutrient supplementation to improve pregnancy outcomes and child health. Dietary interventions starting exclusively during pregnancy have only a little impact on pregnancy outcomes [46]. 

Recently, in The Lancet Global Health, Caniglia et al. compared the protective role of micronutrients—iron only, folic acid only, and iron plus folic acid supplementation—on adverse birth outcomes in 96341 women who gave birth between 2014 and 2020 in Botswana. Micronutrient supplementation included vitamins A, C, D, E, B1, B2, B3, B6, and B12, folic acid, iron, iodine, zinc, selenium, and copper. In this study, it was found that pregnant women who initiated micronutrient supplementation had a lower risk of stillbirth, preterm birth, and low birthweight compared with women who initiated folic acid only, iron only, or folic acid plus iron supplementation [47]. In the future, a mapping of nutrient deficiencies will help clinicians recommend a personalized micronutrient formulation, avoiding the risks of higher concentrations of some micronutrients (dyslipidemia in cases of higher copper concentration, oxidative stress in cases of higher iron concentration, etc.) [48].

### 2.5. Carrier Screening

“Carrier screening” consists of describing genetic testing performed on an asymptomatic individual to determine whether that person has an abnormal allele. Carrier screening can be performed for one specific condition, which includes targeted or extended forms [49]. The likelihood of identifying someone as a carrier for an inherited genetic condition reflects the prevalence of the condition in a particular population. In the absence of a family history, the more prevalent a condition, the greater the likelihood of finding a carrier [50].

Luckily, since genetic testing technology has evolved rapidly over the past decade, it is now possible to screen for a large number of conditions simultaneously. This testing strategy is known as expanded carrier screening. Expanded carrier screening panels typically include options to screen from 5 to 10 conditions, in particular recessive ones, to as many as several hundred conditions [51].

Carrier screening will not identify all individuals who are at increased risk of the screened conditions because it is not possible to screen every disease-producing mutation or allele and because de novo mutations may arise. Patients should be counseled regarding residual risk for every test result. As with all genetic tests, it is important to balance the benefits of screening tests with the risk of emotional and psychological effects on parents. In this scenario, carrier screening should be offered in specific conditions based on familiar medical history and ethnicity [52]. Nowadays, it is possible to screen for different disorders, including hereditary cardiomyopathies, inherited cancer syndromes, lung diseases, and ciliopathies [53]. Genetic counseling is fundamental to defining at-risk couples and interpreting the results of carrier screening tests. 

## 3. Model of Preconception Care in Infertile and Subfertile Couples

Italian regulation recommends that infertile couples undergo the same pre-conception screening as the general population. A diagnostic evaluation is suggested for a woman only after 12 months of unsuccessful regular sexual intercourses in women younger than 35 years old [54]. According to the American College of Obstetricians and Gynecologists (ACOG), women older than 35 years old should receive an evaluation after 6 months of unprotected intercourse [55]. In particular, different studies have demonstrated the great impact of lifestyle factors on fertility [56]:-Weight: for both women and men, the fertility decreases in cases of BMI >25 kg/m^2^ and in cases of BMI <18.5 kg/m^2^ [57]. In detail, obesity is correlated with ovulatory dysfunction and lower birth rates [58]. Couples trying to conceive should aim for a BMI between 18.5 kg/m^2^ and 25 kg/m^2^ [40].-Diet: adherence to a healthy diet based on vegetables, fruits, and fish has a beneficial effect on fertility and live birth rates [59]. In order to avoid periconceptional infection caused by salmonella, toxoplasmosis, or listeria, it is recommended that any vegetables or fruits be washed before eating, that all perishable food be correctly refrigerated, and that meat and eggs be well-cooked before eating [60]. Moreover, the adequate intake of vitamins or mineral supplements, such as iodine, iron, vitamin D, and folic acid, improves fertility and the chances of having a healthy child [61].-Caffeine, smoking, alcohol, or recreational drug abuse: it is recommended to limit the caffeine intake (less than two cups per day) and to avoid smoking, alcohol, and recreational drugs before conception in order to improve fertility and reduce the risk of obstetrical adverse events [62].-Physical activity: it is demonstrated that 150 min of physical activity have beneficial effects on couples trying to conceive, either in the preconception period or during pregnancy [63].

## 4. Model of Preconception Care in Couples with a Previous History of Reproductive Adverse Events

The organization of adequate preconception counseling is an essential step after adverse obstetrical events. 

### 4.1. Previous Stillbirth 

Emotional feelings of vulnerability, depression, and anxiety in post-pregnancy and in the puerperium could be related to the time elapsed since stillbirth. 

Topics of counseling:-To investigate the possible causes according to the consensus of “sudden and unexpected death in fetal life through early childhood”;-To inform the woman that the risk of recurrence of intrauterine death in subsequent pregnancy is generally increased (OR 3.38, 95% CI 2.61–4.38) [64];-To advise the couple of a waiting period of at least 6 months from the intrauterine death event to allow an optimal return to both the physical and psychological integrity of the woman and the couple.

In a recent study, the onset of subsequent pregnancy with a time elapsed from intrauterine death less than 6 months was associated with a three times greater risk of recurrence (OR 3.3); however, when the results were corrected for possibly confounding maternal factors (age, BMI, socioeconomic level, smoking, etc.), the recurrence risk was significantly reduced with an OR of 1.6 [65]. These data are partly confirmed by another study of approximately 15,000 pregnancies with intrauterine deaths in a previous pregnancy, which included 228 intrauterine deaths in a subsequent pregnancy and in which no significant association was found between an interval of less than 6 months and the risk of recurrence [66]. However, it should be noted that this latest study was performed in a population of predominantly Caucasian women with a satisfactory level of social and health care, so the results may not be applicable to the general population. 

The risk of recurrence in subsequent pregnancy depends on the cause of the stillbirth: if the previous intrauterine death was related, for example, to placental vascular diseases, the risk of recurrence was increased compared to the general population (OR 1.96, 95% CI 1.5–3.5); if the cause of the previous stillbirth was related to other causes (infectious, cord accidents, fetal maternal hemorrhage, etc.) or unexplained, the risk of stillbirth in subsequent pregnancy was equal to that of the general population (1.03; 95% CI 0.5–2.2) [67].

There are also nonvascular placental lesions (massive fibrinoid deposits, chronic villus-intervillositis, etc.), which are an expression of possible underlying autoimmune/inflammatory pathologies (including subclinical and unknown to the mother) that have a very high repetition rate (the literature estimates 34% to 100%), with the risk of worsening and anticipation of placental damage at each subsequent pregnancy, from which the clinical consequences derive [68,69]. In this regard, it is therefore essential to better investigate the cause of death, with particular attention to placental histology, in order to better counsel the couple in preparation for the next pregnancy.

### 4.2. Recurrent Pregnancy Loss

According to ESHRE, recurrent pregnancy loss (RPL) is defined as the loss of two or more pregnancies until 24 weeks of gestation, excluding ectopic and molar pregnancies [70]. This condition affects 2–5% of couples trying to conceive [71]. Despite the fact that the pathophysiological mechanisms of RPL still remain unclear, several risk factors have been linked to RPL [72]. Preconception counseling should investigate the possible conditions and risk factors associated with RPL [73]:-Genetic alterations: chromosomal abnormalities, such as aneuploidies. Different genetic tests can be proposed in couples affected by RPL: parental karyotype, products of conception (POC) tests, or pre-implantation testing (PGT) [74].

The parental karyotype has a potential role for a few couples; specifically balanced chromosomal rearrangements are found in 2–5% of RPL couples. In these cases, the cumulative live birth rate and the chances of having a healthy child are good [70]. Moreover, some studies have demonstrated that there is no difference in the live birth rate between couples who undergo PGT and couples who conceive spontaneously [75]. 

The POC tests are POC karyotyping and POC chromosomal microarray. These tests can reveal the aneuploidies, which explain approximately 55% of cases of RPL [76]. In 2018, the ESHRE Group did not recommend the use of POC tests for routine clinical use but only for explanatory purposes. Moreover, in the case of genetic analysis of the pregnancy tissue, it is recommended to use a POC chromosomal microarray [70].

PGT requires in-vitro fertilization and an embryo biopsy. In detail, PGT-Aneuploidies (PGT-A) is developed to screen for aneuploidies [77]. Nowadays, its role is limited to the cases in which aneuploidy has a clinical effect. The ESHRE does not recommend its use if there is a lack of a genetic cause of RPL [70].

-Maternal anatomical alterations: Müllerian anomalies, uterine leiomyoma, and uterine synechiae. Maternal anatomical alterations can explain 15–42% of cases of RPL [78]. Among congenital uterine malformations, the septate uterus can be found in 6–16% of women affected by RPL [79]. Three-dimensional ultrasound is an accurate and noninvasive method for the diagnosis of congenital uterine malformations [80]. Among acquired uterine malformations, the prevalence of leiomyomas in RPL is 0.5–1.3% [79], and the prevalence of synechiae in RPL is 1.3–9.6%, up to 20% after miscarriage [81].-Maternal comorbidities: thrombophilic disorders and endocrinopathies. Investigations should include blood sugar levels, thyroid function tests, and thrombophilia screening [82,83,84,85].

Recent studies found an important correlation between the history of RPL and abnormal glucose metabolism tests [86]. Moreover, there is also an association between an increased risk of miscarriage and thyroid autoimmunity. The prevalence of thyroid autoimmunity in women affected by RPL is 14.8% [87].

The most frequent inherited thrombophilias are factor V Leiden, antithrombin deficiency, prothrombin G20210A, and protein C or protein S deficiency. The role of these mutations in RPL is still debated [88]. Limited studies in patients affected by inherited thrombophilias have not shown evidence of any benefit of treatment with anticoagulants [89,90]. On the contrary, acquired thrombophilias (antiphospholipid antibody syndrome) are strictly associated with RPL [91]. It is estimated that 5–10% of patients affected by RPL screen positive for antiphospholipid antibodies [92]. In these cases, it is recommended to start the treatment with preconception low-dose aspirin (LDA, 75–100 mg daily) and low molecular weight heparin (LMWH, 4000–6000 IU/day) from the moment of the positive pregnancy test [93]. The treatment should be continued for six weeks postpartum to prevent blood clot formation [94]. In detail, the use of aspirin and LMWH reduces the risk of pregnancy loss because of their anticoagulant activity [95]. On the other hand, the risk of other obstetrical complications, such as preeclampsia or premature delivery, is not different in treated patients and in nontreated patients [96]:-Sperm quality: the rate of sperm DNA fragmentation is higher in men within couples affected by RPL [97].-Lifestyle factors: alcohol use, excessive caffeine use, exposure to environmental toxins (organic solvents, mercury, ionizing radiation), smoking, stress, and excessive or lower BMI [98].-Maternal age: the risk of RPL is significantly increased in women older than 40 years old [99]. In 30–34-year-old women, the rate of RPL is 15%; in 35–39-year-old women, the rate of RPL is 25%; in 40–44-year-old women, the rate of RPL is 51%; and in women older than 45 years old, the rate of RPL is 93% [98].-Previous pregnancy loss: in cases of more than five pregnancy losses, the risk of another pregnancy loss is more than 63% [100].

### 4.3. Preterm Birth

According to the World Health Organization (WHO), preterm birth (PTB) is defined as birth before 37 weeks of gestation [101]. Although the underlying causes are largely unknown, some risk factors have been identified [102]. First, Miller et al. showed that in these couples, the risk of recurrence of preterm birth is generally increased (RR 2.62, 95% CI 1.99–3.44) [103]. Adequate preconception counseling is important for couples who have experienced prior PTB. Additionally, some baseline patient characteristics are linked to PTB: low socioeconomic status, non-Hispanic Black race, underweight pre-pregnancy BMI, maternal smoking, cocaine or opioid abuse, and family history of PTB [104]. Moreover, intrauterine infection, multiple gestations, congenital uterine malformation, and shortened cervical length (less than 25 mm) are identified as PTB risk factors [105]. 

Preconception counseling is fundamental to recommending preventive treatments for at-risk patients [106]:-Progesterone supplementation: several studies demonstrated that women with a history of PTB who received progesterone had a significant reduction in the rate of PTB [107]. Nowadays, progesterone supplementation has become the standard of care for women at high risk of PTB [108]. Additionally, progesterone has been shown to also reduce the risk of PTB in women who have a cervical length less than 25 mm in the mid-trimester [109]. On the other hand, some researchers found that the use of progesterone has no benefit in reducing the PTB rate in at-risk women [110]. Further studies are needed to understand the potential role of progesterone in these women.-Cervical cerclage: this is a surgical procedure in which sutures reinforce the cervix. In women who experienced a prior PTB < 34 weeks, prophylactic cerclage may be offered. Specifically, some researchers found that cerclage reduces the risk of birth before 35 weeks in women with prior PTB and with a cervical length less than 15 mm [111].-Specialized clinics: at-risk women treated in specialized prematurity clinics have better obstetrical outcomes. The implementation of prematurity programs is strictly linked to reductions in the rates of recurrent PTB in these patients [107].

## 5. Conclusions

Preconception care is a potential resource for future parents, specifically for those who have modifiable risk factors. Preconception care should occur any time any gynecologist/andrologist/psychologist/geneticist/nutritionist or other health care provider meets a couple who is considering planning a pregnancy. 

The multidisciplinary team guarantees synergy, which turns out to be “direct” counseling. A shared medical record would help different specialists monitor and follow up on women from preconception to puerperium. 

## Figures and Tables

**Table 1 jcm-12-06352-t001:** Topics to be discussed in a preconception visit in a couple without a previous history of infertility or previous adverse reproductive events.

First Preconception Visit	
Reason for the visit	Preconception
Family history	Family tree
Personal history	General health status Chronic disorders Genetically dominant diseases Nutritional status and BMI Allergies Work exposure Smoking habits, alcohol, and drug use Psychiatric disorders Dental hygiene
Andrological assessment	For men
Supplementation	Folic acid: 400 micrograms
Carrier screening	Based on family history
Vaccinations	SARS-CoV-2 Influenza during the flu season Varicella and rubella Tetanus/diphtheria/pertussis
Infection screening	Rubella titer and varicella titer HIV testing and hepatitis C antibody Cervical cytology

## Data Availability

No new data were created in this study.

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
