# Peer review of "Before Is Better: Innovative Multidisciplinary Preconception Care in Different Clinical Contexts"

_jcm, 2023, doi:10.3390/jcm12196352_

Round 1

Reviewer 1 Report

Dear

Your review is very interesting but I have some recommends:

In patients whose pregnancies were complicated by recurrent pregnancy loss, you can use screening for thrombophilia. When it is positive and findings show the risk for adverse perinatal outcomes You can implement anticoagulation therapy in preconception time. Would You like to give us some information about it in this review as well as about the possible results of preconception  anticoagulant therapy to perinatal outcomes in pregnancy complicated with congenital thrombophilia.

Minor editing of English language required

Author Response

Thank you very much for taking the time to review this manuscript. In the re-submitted manuscript I add detailed information about the therapy in patients affected by thrombophilia. You can find the corresponding revision highlighted in the file.  

Reviewer 2 Report

This manuscript provides models of preconception care  for the general population, for infertile/subfertile population and for couples with history of adverse reproductive events. Elements of the models are carefully selected, well-structured and can help to implement early preventive, diagnostic and therapeutic measures. The multidisciplinary approach is of outmost importance to improve the preconception care. Each item of the models is supported by relevant references.

Acknowledging the pratical value of the manuscript the reviewer has some concerns to be considered: 1/ In developed countries high-standard preconception care is operating with reasonable success. 2/ The preconception care applied in these country incorporates factors proposed by the authors' models. 3/ The proposed models therefore have modest scientific value, but it can be regarded as an important public health issue.

Minor comments: The references should be checked, there are some inconsistences in the use of "et al" and in the "abbreviations of journal names".

Author Response

Thank you very much for taking the time to review this manuscript.

1) To answer to your concerns I want to start citing the first sentences in the article "It is well-known that preconception interventions, and specifically the promotion of optimal preconception maternal and paternal health status, reduce the risk of adverse reproductive events. Despite this, in Italy preconception counselling is still overlooked." In many developed countries the preconception care is applied successfully, but there are other developed countries, such as Italy, in which the preconception care is still under evaluated. For example, as I wrote in the article, only 20% of Italian women started folic acid supplementation before pregnancy.

2) In our review we tried to create a model which focus on the principal points of an efficient preconception care

3) We agree with you! We believe in multidisciplinary approach. Our review should provide not only gynecologists with guidelines on preconception care. Andrologists, psychologists, geneticists, nutritionists or other health care providers meet a couple who is considering planning a pregnancy are involved. In this scenario, our model can be important in public health issue.

We checked the references. Please find the corrections in red in the re-submitted manuscript.